# Genetic Diversity and Dispersal of DENGUE Virus among Three Main Island Groups of the Philippines during 2015–2017

**DOI:** 10.3390/v15051079

**Published:** 2023-04-28

**Authors:** Ava Kristy Sy, Carmen Koo, Kristine J. R. Privaldos, Mary Ann T. Quinones, Mary A. U. Igoy, Sharon Y. A. M. Villanueva, Martin L. Hibberd, Lee Ching Ng, Hapuarachchige C. Hapuarachchi

**Affiliations:** 1National Reference Laboratory for Dengue and Other Arbovirus, Virology Department, Research Institute for Tropical Medicine, Filinvest Corporate City Compound, Alabang, Muntinlupa City 1781, Philippines; avakristysy@gmail.com (A.K.S.); privaldoskristinejoy@gmail.com (K.J.R.P.); meannquinones@gmail.com (M.A.T.Q.); meannigoy20@gmail.com (M.A.U.I.); 2Environmental Health Institute, National Environment Agency, 11, Biopolis Way, #06-05-08, Singapore 138667, Singapore; carmen_koo@nea.gov.sg (C.K.); ng_lee_ching@nea.gov.sg (L.C.N.); 3Department of Medical Microbiology, College of Public Health, University of the Philippines Manila, 625, Pedro Gil Street, Ermita, Manila 1000, Philippines; smvillanueva4@up.edu.ph; 4Department of Infection Biology, Faculty of Infectious & Tropical Diseases, London School of Hygiene & Tropical Medicine, Keppel Street, London WC1E 7HT, UK; martin.hibberd@lshtm.ac.uk; 5Philippine Genome Centre, Dilliman Campus, University of the Philippines, Dilman, Ma. Regidor, U.P. Campus, Quezon City 1101, Philippines; 6National Institutes of Health, University of the Philippines Manila, 623, Pedro Gil Street, Ermita, Manila 1000, Philippines; 7Genome Institute of Singapore, 60, Biopolis Street, Genome, #02-01, Singapore 138672, Singapore; 8School of Biological Sciences, Nanyang Technological University, Singapore 639798, Singapore

**Keywords:** dengue virus, evolution, molecular epidemiology, genetic diversity, phylogeography

## Abstract

Dengue has been one of the major public health concerns in the Philippines for more than a century. The annual dengue case burden has been increasing in recent years, exceeding 200,000 in 2015 and 2019. However, there is limited information on the molecular epidemiology of dengue in the Philippines. We, therefore, conducted a study to understand the genetic composition and dispersal of DENV in the Philippines from 2015 to 2017 under UNITEDengue. Our analyses included 377 envelope (*E*) gene sequences of all 4 serotypes obtained from infections in 3 main island groups (Luzon, Visayas, and Mindanao) of the Philippines. The findings showed that the overall diversity of DENV was generally low. DENV-1 was relatively more diverse than the other serotypes. Virus dispersal was evident among the three main island groups, but each island group demonstrated a distinct genotype composition. These observations suggested that the intensity of virus dispersal was not substantive enough to maintain a uniform heterogeneity among island groups so that each island group behaved as an independent epidemiological unit. The analyses suggested Luzon as one of the major sources of DENV emergence and CAR, Calabarzon, and CARAGA as important hubs of virus dispersal in the Philippines. Our findings highlight the importance of virus surveillance and molecular epidemiological analyses to gain deep insights into virus diversity, lineage dominance, and dispersal patterns that could assist in understanding the epidemiology and transmission risk of dengue in endemic regions.

## 1. Introduction

Dengue has been one of the major public health problems in Asia for centuries. Among the six WHO regions, most outbreaks reported in the literature from 1990 to 2015 have occurred in Southeast Asia and the Western Pacific (WPRO) regions [1]. The number of annual dengue burdens more than doubled from 2013 to 2019 in WPRO, surpassing 1 million cases in 2019 [2]. The Philippines has been one of the major dengue-endemic countries in WPRO since its first dengue outbreak was documented in 1906 among American military soldiers [3]. The first recorded epidemic of severe dengue in Southeast Asia occurred in Manila in 1954 [4,5] due to DENV-3 [6]. Subsequent epidemics occurred in 1966, 1983, 1998, and 2001 with an increasing magnitude [6,7,8]. Multiple DENV serotypes were present during these epidemics and more than one serotype dominated in each episode [6,8]. The epidemic in 1998 due to DENV-2 was one of the worst in the Philippines’ history, recording the highest incidence rate (60.9 cases per 100,000 population) and case fatality rate (2.6%) [6,9]. Of approximately 1.56 million cases reported in WPRO from 2013 to 2019, 42.6% of cases were from the Philippines [2]. Moreover, 75.6% of the dengue-related mortality in WPRO has also been reported from the Philippines during the same period [2], indicating a substantial public health burden due to dengue in the country among other countries/territories in the Asia subregion of WPRO. The rising trend of dengue in the Philippines has been facilitated by multiple factors, such as rapid urbanization, poor sanitary practices, and improper waste disposal management [10].

Since the early 1990s, plenty of molecular epidemiological studies have been conducted to determine the genetic diversity, evolution, and emergence of new DENV strains in endemic areas [11,12,13,14,15]. The information gathered through those studies has been useful to evaluate vaccine efficacy, deciphering determinants of disease severity, and to predict the magnitude of impending outbreaks [16,17,18,19,20]. Despite being hyperendemic to dengue, there is only a handful of reports from the Philippines that narrate a comprehensive molecular epidemiological picture of circulating DENV populations [21,22]. Such analyses have been hindered by limitations in virus genomic data, both in time and space, due to inconsistent sampling as well as inadequate laboratory and technical capacity [23,24]. In this present study, we aimed to fill this knowledge gap by systematic sampling from 21 sentinel sites between 2015 and 2017 and analyzing envelope (*E*) gene sequences (*n* = 377) of the 4 DENV serotypes. We determined the evolutionary history, fluctuations in genetic diversity, and virus dispersal patterns in different regions of the Philippines. Our findings further the understanding of the molecular epidemiology of DENV in the Philippines, which is pivotal for the assessments of dengue epidemic risk, transmission patterns, and vaccine efficacy.

## 2. Materials and Methods

### 2.1. Study

This present study was facilitated by UNITEDengue “www.unitedengue.org (accessed on 1 March 2023)”, a network for the cross-border surveillance and sharing of data as well as expertise on dengue. The two collaborative institutes, the Environmental Health Institute, National Environment Agency, Singapore, and Research Institute for Tropical Medicine, Ministry of Health, Philippines, are members of UNITEDengue.

### 2.2. Sample Collection

Epidemiological data were collected through the Philippines Integrated Disease Surveillance and Response (PIDSR). This is a national surveillance program of diseases categorized as notifiable to the Department of Health. Dengue is included as one of the notifiable diseases and disease reporting units (DRU) are required to report cases on a weekly basis. Basic epidemiological data were collected from suspected dengue cases, including age, sex, date of symptom onset, date of reporting, patient/DRU address (barangay, municipality, province, region), symptoms, and outcome. Suspected dengue cases were defined as those with a self-reported sudden acute fever and at least two additional warning signs as per 2009 World Health Organization (WHO) guidelines [25]. Dengue was confirmed by detecting DENV *RNA* in suspected samples as per the protocol described below for the serotyping of the virus.

Virological surveillance was initiated in 2014 to collect serum samples from dengue-suspected patients visiting selected DRUs and hospitals. A total of 5 randomly selected samples from 21 sentinel hospitals and DRUs with a marked increase in suspected dengue cases were included in the virological surveillance every week. Samples were sent to the Research Institute of Tropical Medicine for serotyping and genomic analysis.

### 2.3. Serotyping of Dengue Virus

DENV *RNA* was extracted from serum samples sent to the Research Institute for Tropical Medicine by using QIAamp^®^ Viral *RNA* Mini Kit (Qiagen GmbH, Hilden, Germany). DENV was typed by using multiplex single-step real-time reverse transcription polymerase chain reaction (RT-PCR), as described previously [26].

Briefly, 5 μL of *RNA* template was used in a total reaction of 25 µL containing 50 pmol of DENV-1 and DENV-3 primers, 25 pmol of DENV-2 and DENV-4 primers, 9 pmol of serotype-specific probes, and 5 uL of SuperScript™III Reverse Transcriptase (ThermoFisher, Waltham, MA, USA). The amplification protocol included reverse transcription at 50 °C for 30 min, followed by PCR with initial denaturation at 95 °C for 2 min, 45 cycles of denaturation at 95 °C for 15 s, and annealing at 60 °C for 1 min.

### 2.4. Reverse Transcription-Polymerase Chain Reaction Amplification and Sequencing of Envelope Gene of Dengue Virus

The synthesis of complementary *DNA* (*cDNA*) from *RNA* extracted from sera was carried out using The Maxima H Minus First Strand *cDNA* Synthesis Kit (Thermo Fisher Scientific Inc, Waltham, MA, USA) according to manufacturer’s recommendations. The amplification of complete *E* gene (~1.5 kb) was conducted in a 20 μL reaction mixture using 0.5 uM of DENV serotype-specific primers (Appendix A) and 1X PhusionTM Flash High-Fidelity PCR Master Mix (ThermoFisher Scientific, Waltham, MA, USA). The cycling condition was 10 s denaturation at 98 °C, followed by 35 cycles of 5 s denaturation at 98 °C, 10 s annealing at 62 °C, and 45 s extension at 72 °C, ending with a final extension for 1 min at 72 °C. Amplified PCR products were visualized in 1.5% agarose gels stained with GelRed (Biotium Inc., Fremont, CA, USA). PCR products were purified using Expin PCR SV mini kit (GeneAll Biotechnology, Seoul, Republic of Korea) and then sequenced using DENV serotype-specific sequencing primers (Appendix A) at a commercial facility according to the BigDye Terminator Cycle Sequencing kit (Applied Biosystems, Waltham, MA, USA) protocol.

### 2.5. Assembly and Comparison of Nucleotide Sequences

Overlapping raw nucleotide sequences were assembled to generate the consensus sequences of *E* gene for individual isolates using the Lasergene version 8.0 (DNASTAR Inc. Madison, WI, USA). Alignment of contiguous sequences and visual inspection of alignment were performed using BioEdit 7.0.5 software suite [27]. The *E* gene sequences of the study isolates were compared with those available in GenBank database to determine sequence similarity and to identify unique substitutions in study isolates.

### 2.6. Selection Pressure Analysis

Envelope gene sequences of study isolates (*n* = 377; DENV-1 = 118, DENV-2 = 71, DENV-3 = 139, and DENV-4 = 49) and global reference sequences retrieved from GenBank database (*n* = 765; DENV-1 = 273, DENV-2 = 191, DENV-3 = 175 and DENV-4 = 126) were analyzed using HyPhy 2.5 software package implemented in the Datamonkey webserver [28], as described previously [29]. The datasets for each serotype were analyzed separately. Single likelihood ancestor counting (SLAC), fixed effects likelihood (FEL), internal fixed effect likelihood (IFEL), mixed effect model of evolution (MEME), and fast unbiased Bayesian approximation (FUBAR) methods were used to determine whether any amino acid residue was under selection pressure. The non-neutral residues (positively or negatively selected sites) were confirmed by at least three methods (significant levels were defined as posterior probability value of ≥0.9 for FUBAR and *p*-values ≤ 0.05 for the remaining methods).

### 2.7. Phylogenetic Analysis of Envelope Gene Sequences

Time-scaled phylogeny construction was performed based on the Markov Chain Monte Carlo (MCMC) method available in Bayesian Evolutionary Analysis by Sampling Trees (BEAST) software package v. 1.7.4 [30]. The dataset included the representative DENV *E* gene sequences of 277 study isolates (DENV-1 = 89, DENV-2 = 55, DENV-3 = 106, and DENV-4 = 27) and 334 reference sequences (DENV-1 = 126, DENV-2 = 77, DENV-3 = 69 and DENV-4 = 62) obtained from GenBank database. This subset included at least one sequence representative of a group of identical sequences from each region per epidemiological week. The analysis was performed for each serotype separately. Tamura-Nei model with gamma correction (TN93+G) was selected as the best-fit model via jModel Test [31]. In order not to assume any demographic scenario as a priori, a relaxed uncorrelated lognormal clock [32] and the Bayesian skyline plot (BSP) coalescent model [33] were used to estimate the most probable origin, nucleotide substitution rate, and the time to most recent common ancestor (tMRCA) of different DENV lineages. The prior substitution rate was set at default settings to allow estimation of the rates among tree branches. The MCMC chain was run for 200–500 million generations sampling every 20,000–50,000 states, respectively. Effective Sampling Size (ESS) value of more than 200, visualized via Trace v. 1.5 [34] program, was considered a sufficient level of convergence of parameters. The posterior tree distribution was summarized via Tree Annotator v. 1.7.4 program [30], with 10% burn-in, and the final MCC tree was visualized via FigTree v. 1.4.4 “http://tree.bio.ed.ac.uk/software/figtree/ (accessed on 1 March 2023)”.

### 2.8. Phylogeography Analysis of Envelope Gene Sequences

We inferred dispersal patterns of DENV by using the complete set of *E* gene sequences (*n* = 377) and a discrete phylogeography approach. Sequences were categorized into 17 discrete states based on case locations in 17 administrative regions. The analysis was repeated considering 3 main island groups (Luzon, Visayas, and Mindanao) as discrete states (*n* = 3) to confirm the validity of spatial diffusion pathways inferred based on 17 regions. The latitude and longitude of the center of each region/island group were used to define the location of each discrete state. Analyses were carried out via BEAST version 1.7.4 [30], with Tamura-Nei model with gamma correction (TN93+G) substitution model, BSP coalescent model, and uncorrelated log-normal relaxed molecular clock model. A Bayesian Stochastic Search Variable Selection (BSSVS) procedure was used to identify significant dispersal pathways among 17 discrete states. MCMC chain was run until convergence (DENV-1 and DENV-3 = 200 million iterations, DENV-2 = 150 million iterations, DENV-4 = 100 million iterations), sampling every 10,000 states. The convergence of the parameters was assessed as described in the phylogenetic analysis. TreeAnnotator v. 1.7.4 [30] was used to summarize the MCC tree. The root state posterior probability value of each discrete location was extracted from the annotated MCC tree using FigTree v. 1.4.4 “http://tree.bio.ed.ac.uk/software/figtree/ (accessed on 1 March 2023)” to determine the most probable location of virus lineage origin. SPREAD 1.0.4 [35] was used to analyze phylogeographic reconstructions resulting from the Bayesian inference of spatiotemporal diffusion of tests established. Bayes factor (BF) values were calculated by comparing the posterior and prior probability of individual rates to test the significant linkage between locations. BF > 3 was considered well supported, with sub-classifications of substantial (BF > 3), strong (BF > 10), very strong (BF > 30), and decisive (BF > 100) support [33,36].

## 3. Results

### 3.1. Dengue Is Widespread and Hyperendemic in the Philippines

The Philippines is an archipelago of over 7000 islands in the Western Pacific Ocean extending over an approximately 300,000 km^2^. It consists of 3 main geographical divisions (island groups) split into 17 administrative regions, namely Luzon [Regions I, II, III, IVA, V, XIV (National Capital Region; NCR), XV (Cordillera Administrative Region; CAR), and XVII], Visayas (Regions VI, VII, and VIII) and Mindanao [Regions IX, X, XI, XII, XIII, and XVI (Autonomous Region in Muslim Mindanao; ARMM)]. Based on the 2015 census, the Philippines is the 12th most populated country with more than 100 million individuals. Approximately 40% of the population lives in 3 regions: Calabarzon (Region IVA; 12.61 million), NCR (11.86 million), and Central Luzon (Region III; 10.14 million). The Philippines has a tropical climate with dry (December to May) and wet (June to December) seasons, and the average annual temperature ranges between 27 °C and 37 °C. Dengue cases generally peak during the monsoon season (January to November), when the environment becomes conducive for *Aedes* mosquito breeding. A previous study has shown a strong correlation between the rainfall pattern and dengue incidence in Metro Manila [37].

The Philippines reported an average of 217,000 dengue cases in 2015 and 2016, which was an almost 2-fold increase compared to 113,485 cases recorded in 2014 [2]. However, the total number of dengue cases dropped to 152,224 in 2017, recording a 30% lower number of cases than in 2016 (220,518 cases) [2]. Most dengue cases reported during this study period (2015–2017) occurred in Luzon Island (Figure 1). In 2016, the dengue case burden was substantial also in the other two main island groups (Visayas and Mindanao). The high case burden in Luzon Island is not surprising because certain areas, such as Central Luzon, Calabarzon, and NCR, are densely populated and have witnessed large epidemics in the past [4,5,7,8]. In addition, the % CFR increased substantially during 2016–2017 compared to that during 2014–2015. The highest % CFR was in the age group below 10 years (0–4 years old = 0.80%; 5–9 years old = 0.91%), suggesting that dengue remains an important pediatric illness in the Philippines [38]. We observed all four DENV serotypes in the study sample cohort, indicating that dengue is widespread and hyperendemic in the Philippines (Figure 1).

### 3.2. DENV Serotypes Consisted of Multiple Clades That Constituted Distinct Genotype Compositions in Each Island Group

Our analyses included 377 *E* gene sequences (minimum length of 1395 bp) belonging to DENV-1 (*n* = 118), DENV-2 (*n* = 71), DENV-3 (*n* = 139), and DENV-4 (*n* = 49). Isolates sequenced were from all three main island groups, but not with equal contribution from each island group. The majority were from Luzon (70.6%), followed by Mindanao (20.1%) and Visayas (9.3%). The variation in sample size in each island group is due to the different capacities of the regional surveillance systems and the population size of the hospital catchment areas.

We estimated the population composition, origin, ancestral age, and evolutionary dynamics of DENV lineages by using a Bayesian phylogenetic approach. For those analyses, we selected 277 *E* gene sequences (89 DENV-1, 55 DENV-2, 106 DENV-3, and 27 DENV-4) from the total dataset of 377 sequences. This subset was selected to minimize redundancy in the analyses and included at least one sequence to represent a group of identical sequences from each region reported during each epidemiological week. The dataset also included 334 *E* gene sequences (126 DENV-1, 77 DENV-2, 69 DENV-3, and 62 DENV-4) retrieved from the Genbank database for comparison. The study sequences showed low genotype diversity across all serotypes. Each serotype consisted of a single genotype, except for DENV-1, which included two genotypes (Appendix A). This was even though circulating serotypes were not newly emerged in the country, according to the ancestral age analyses of Philippine strains (Appendix A).

However, the dominant genotypes of DENV-1 (genotype II), DENV-2 (cosmopolitan genotype), DENV-3 (genotype I), and DENV-4 (genotype II) formed 12 distinct clades supported by strong posterior probability (0.8–1.0). These clades differed in genetic similarity, mean evolutionary rate, ancestral age, and the period of transmission (Table 1). All clades had basal/outgroup sequences from regional countries, suggesting their Asian ancestry (Appendix A). However, each clade largely included local sequences and showed minimal mixing with sequences reported from regional countries, suggesting the presence of distinct virus lineages in the Philippines. These clades, except for those of DENV-1, possessed unique and novel amino acid substitutions in the *E* gene, which were either neutral or under purifying selection (Appendix A). The ancestral age analyses indicated that clades with longer periods of transmission emerged earlier than others, allowing them to evolve further and become genetically distinct (Table 1). Among 12 clades, the majority were dominant in Luzon (*n* = 6) and Mindanao (*n* = 5), except for the DENV-3 genotype Ic, which was mainly found in Visayas. Even though the mixing of sequences from different island groups was evident for each clade, the overall genotype composition in each island was distinct (Figure 2). This was most obvious between Luzon and Mindanao, which are northern and southern island groups, respectively. Visayas showed a composition similar to that in Luzon, but this needs to be interpreted with caution given the small number of analyzed sequences from Visayas.

### 3.3. Luzon Island Is the Most Probable Hub of DENV Dispersal in the Philippines

In agreement with the phylogenetic analyses, Bayesian phylogeography analyses based on 377 *E* gene sequences of all 4 DENV serotypes also demonstrated the mixing of virus strains in different regions of the Philippines (Figure 3). Nonetheless, relatively high spatial clustering of all four serotypes was present in Luzon and Mindanao (Appendix A), similar to that observed in the clade distribution pattern (Figure 1). The Bayesian phylogeography analyses identified 34 well-supported dispersal pathways [Bayes Factor (BF) = 3–425; Appendix A among 4 serotypes (DENV-1 = 10, DENV-2 = 9, DENV-3 = 11, and DENV-4 = 4). The number of pathways for each region is summarized in Table 2. As detailed in Appendix A, the majority (61.8%; 21/34) of pathways were connections within the same island group, while the remaining pathways (38.2%; 13/34) connected different island groups. Out of 13 significant island-to-island connections, there were 8 location pairs with Visayas connected to the other 2 island groups, Luzon and Mindanao, and the majority (87.5%; 7/8) of these pathways were between Luzon and Visayas. This further supported our previous observation on the higher similarity of genetic composition between these two island groups (Luzon and Visayas).

Based on the highest root state probability values, Calabarzon was the most probable source population for DENV-2 and DENV-3, while CAR and CARAGA were the most probable source populations for DENV-1 and DENV-4, respectively (Figure 4). CAR also showed comparable root state probability values for DENV-4. Moreover, NCR also demonstrated consistently high root state probability values for DENV-1, -2, and -3. Calabarzon and CARAGA harbored the highest number of lineages (*n* = 10) among all regions analyzed, indicating the presence of genetically diverse DENV populations in these two regions. Based on these observations, we suggest that Luzon Island could serve as one of the major sources of DENV dispersal in the Philippines. Besides, Mindanao is also likely to be an efficient hub of DENV-4 dispersal.

## 4. Discussion

Dengue fever is hyperendemic in the Philippines, where all four serotypes of DENV are circulating. In the last decade, dengue has been an increasing trend in the country, especially since 2015, and recorded the highest-ever case burden of 437,563 cases in 2019 [2]. The lack of stakeholder empowerment, difficulty in eradicating local breeding habitats, and less efficient solid waste management have been identified as several contributory factors to the recent surge in cases [39], undermining efforts of the National Dengue Prevention and Control Program established in 1993 [40]. Despite its historical presence and growing public health importance, the number of scientific publications on dengue in the Philippines has been strikingly low [21] since dengue became a notifiable disease in the country in 1958 [40]. The scarcity of information is especially noticeable regarding dengue virology and molecular epidemiology [21]. This present study aimed to address this concern by investigating the composition, evolutionary dynamics, and spread of DENV lineages in the Philippines from 2015 to 2017. To the best of our knowledge, this is the first study that describes the dispersal pattern/s of DENV among three island groups of the Philippines using phylogeographic analyses.

The genotype diversity of each DENV serotype in the Philippines was relatively low during the study period. Each serotype consisted of a single genotype, except for DENV-1, which included two genotypes. However, the genetic diversity was more pronounced within each genotype and included 12 distinct clades among 4 serotypes. This observation is characteristic of DENV epidemiology in which a single genotype generally dominates within each serotype in a cyclical pattern [41,42], yet high genetic diversity is maintained within genotypes through co-circulating clades and in situ evolution [43]. A similar pattern has also been reported in previous studies in the Philippines [22,44]. The genotype findings of this present study agree with the observations of previous analyses of DENV sequences in the Philippines from 1956 to 2017 [22,45], except that two genotypes of DENV-1 (genotypes I and II) were detected in this present study. However, genotype I of DENV-1 was detected at an extremely low level (4.2%) compared to that of genotype II among 118 DENV-1 samples analyzed herein, suggesting the overall dominance of genotype II in DENV-1. Nevertheless, the findings highlight the need for thorough surveillance to capture less-dominant genotypes that may cause future genotype shifts associated with outbreaks [46].

It was noteworthy that each Philippine clade of DENV demonstrated minimal representation from regional countries based on available sequence data. Together with the ancestral age data of up to 13 years among the dominant lineages, findings suggested the long-term circulation of distinct virus lineages in the Philippines. Even though these clades, except for those of DENV-1, possessed unique amino acid substitutions, there was no evidence of positive selection among *E* gene sequences. This may suggest that the evolutionary dynamics of these clades were not largely adaptive for fitness gains either in human hosts or local vectors [47], despite their long existence and association with outbreaks. Even though the accuracy of selection pressure analyses could have been improved if complete genomes of DENV were available, the observations based on *E* gene sequences agree with the empirical knowledge on DENV evolution in which purifying selection is more evident than positive selection [48,49], with a notable contribution by genetic drift to the overall genetic diversity [50].

Both genotyping and phylogeographic analyses demonstrated long-range virus dispersal among three island groups. However, the evidence was stronger between island groups located close by, such as Luzon and Visayas, than those located further away, such as Luzon and Mindanao. This is likely due to geographical distance and variable human movement among island groups because the long-range virus dispersal is more likely to be driven by humans than *Aedes aegypti* vectors that have a short flight range of approximately 100–300 m [51]. Nevertheless, approximately 62% of dispersal connections were within the same island group and the overall genotype composition in each island group was distinguishable, suggesting distinct molecular epidemiological patterns in each island group. Phylogeographic analyses indicated Luzon Island as the major dispersal hub of all four serotypes of DENV and Mindanao for DENV-4 in the Philippines. This is not surprising because highly urbanized and populated cities, such as Manila, are in Luzon. Dengue burden has historically been the highest in most populated urban areas such as NCR, where Manila and other urbanized centers are located [24]. Manila has been hyperendemic to dengue for many decades and is well connected internationally and domestically through sea and air.

## 5. Conclusions

Despite having a low genotype complexity, the DENV population circulated in the Philippines from 2015 to 2017 consisted of genetically distinct clades that were variably distributed among three island groups. Luzon and Mindanao were the major hubs of virus dispersal, forming an extensive network of dispersal pathways. Though there was ample evidence of virus migration, the relatively strong spatial structure of virus lineages in island groups, especially Luzon and Mindanao, suggested that geographical separation plays an important role in the evolutionary landscape of the indigenous DENV population. These findings highlight the importance of virus surveillance and molecular epidemiological analyses to gain deep insights into virus diversity, lineage dominance, and dispersal patterns that assist in understanding variations in vaccine efficacy and transmission risk in dengue-endemic regions.

## Figures and Tables

**Figure 1 viruses-15-01079-f001:**
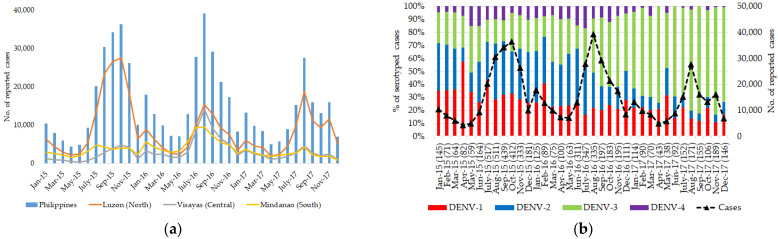
Dengue cases and DENV serotypes in the Philippines from 2015 to 2017: (**a**) monthly distribution of cases in the Philippines (overall) and three main island groups; (**b**) monthly proportion of DENV serotypes in the Philippines.

**Figure 2 viruses-15-01079-f002:**
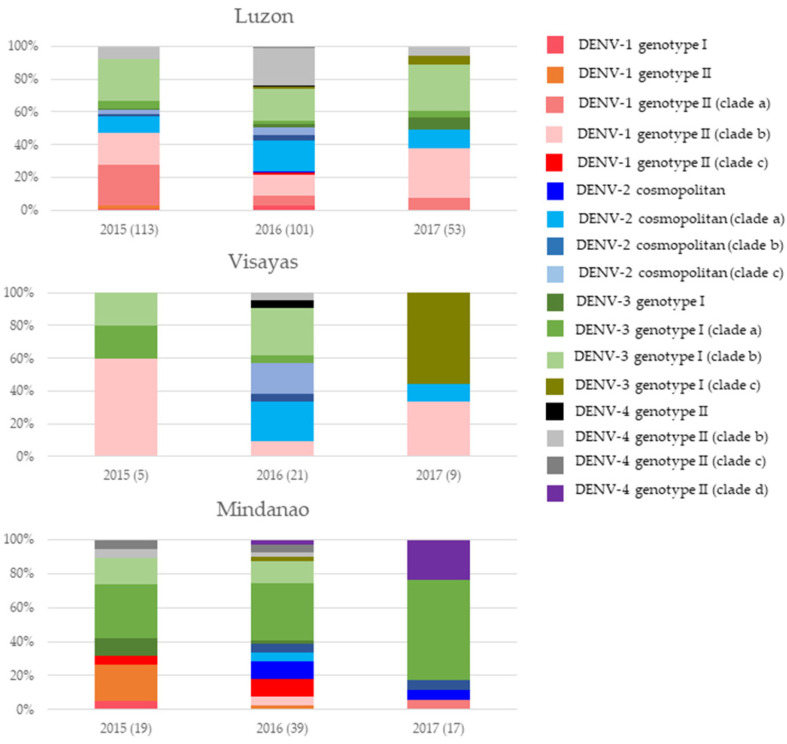
The genotype composition of DENV serotypes in three island groups, namely Luzon, Visayas, and Mindanao. The nomenclature of clades is as per the phylogenetic analyses illustrated in Appendix A. Each color represents a genetically distinguishable clade as named in the legend. Numbers in parentheses indicate the number of sequences analyzed.

**Figure 3 viruses-15-01079-f003:**
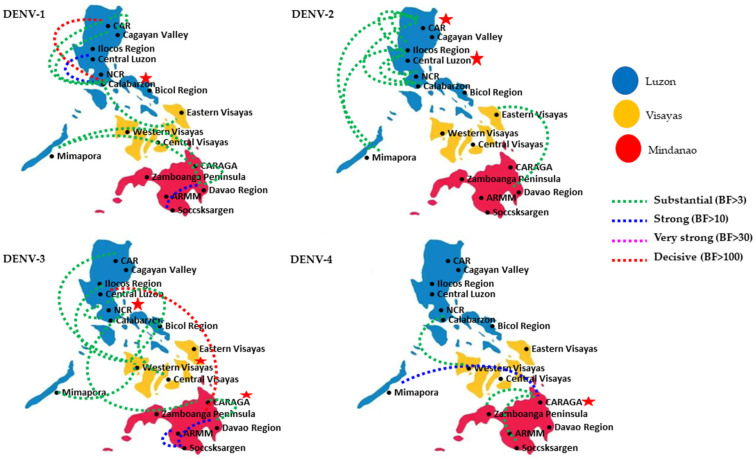
The network of DENV dispersal among different regions in the Philippines. The dispersal patterns were inferred by using the Bayesian Stochastic Search Variable Selection (BSSVS) procedure in BEAST 1.7.4 [30]. The analysis included 377 *E* gene sequences reported from 17 administrative regions. Sequences were categorized into 17 discrete states based on case locations in 17 administrative regions. The latitude and longitude of the center of each region were used to define the geo-position of each discrete state. Any link supported by Bayes factor (BF) >3 was considered significant. Only significant links are shown in the figures. The branch color indicates the BF values (highest in red and lowest in green). The BF values of each link are given in Appendix A. The regions marked with red stars are the locations with maximum number of links.

**Figure 4 viruses-15-01079-f004:**
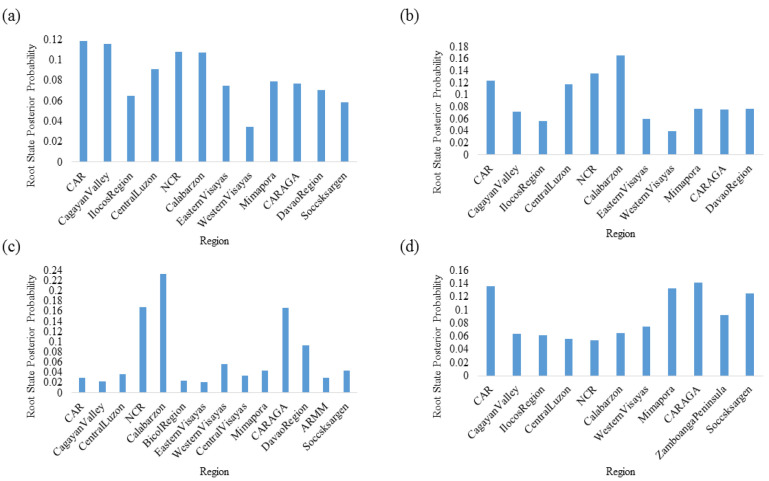
Root state posterior probability analysis for each serotype. The root state posterior probability value of each discrete location (region) was extracted from the annotated maximum clade credibility tree for (**a**) DENV-1, (**b**) DENV-2, (**c**) DENV-3, and (**d**) DENV-4 via FigTree v. 1.4.4 “http://tree.bio.ed.ac.uk/software/figtree/ (accessed on 1 March 2023)” to determine the most probable location of origin of virus lineages.

**Table 1 viruses-15-01079-t001:** Summary of evolutionary analyses of dominant lineages among four DENV serotypes.

Serotype	DENV Strains	Genetic Similarity (Nucleotide and Amino Acid Level)	Mean Evolutionary Rate (95% HPD)	tMRCA (95% HPD)	Transmission Period
DENV-1	Genotype IIa	99.1–100% (98.9–100%)	8.6 (3.7–14.3)	4.8 (3.6–6.6)	June 2015–May 2017
	Genotype IIb	97.7–100% (98.9–100%)	10.3 (4.8–17.0)	13.3 (10.7–16.4)	January 2015–May 2017
	Genotype IIc	99.5–99.9% (99.9–100%)	5.4 (2.2–9.1)	4.0 (2.7–7.3)	November 2015–November 2016
DENV-2	Cosmopolitan genotype a	98–99.9% (98.5–100%)	12.5 (4.2–22.9)	8.4 (6.4–10.7)	March 2015–March 2017
	Cosmopolitan genotype b	99.0–100% (99.1–100%)	10.3 (4.6–17.6)	5.2 (3.5–7.2)	July 2015–April 2017
	Cosmopolitan genotype c	98.6–100% (98.9–100%)	8.5 (2.2–16.7)	7.1 (5.0–9.7)	August 2015–August 2016
DENV-3	Genotype Ia	98.4–100% (98.9–100%)	8.5 (2.1–17.8)	8.4 (4.4–9.0)	June 2015–April 2017
	Genotype Ib	98–100% (99.1–100%)	7.9 (1.8–16.6)	6.4 (5.7–11.7)	March 2015–May 2017
	Genotype Ic	98.7–99.9% (99.5–100%)	6.2 (1.4–12.5)	5.4 (3.1–8.3)	September 2016–April 2017
DENV-4	Genotype IIb	98.5–100% (99.1–100%)	8.0 (1.1–17.7)	7.0 (4.9–9.7)	August 2015–February 2017
	Genotype IIc	99.5–99.9% (99.1–100%)	11.1 (2.4–23.7)	3.7 (2.2–5.9)	November 2015–December 2016
	Genotype IId	99.2–99.7% (99.3–100%)	4.5 (1.6–8.0)	3 (1.2–5.7)	September 2016–May 2017

**Table 2 viruses-15-01079-t002:** The number of dispersal pathways for virus lineages in each region of different island groups.

Island Group	Region	No. of Dispersal Pathways ^£^			
		DENV-1	DENV-2	DENV-3	DENV-4
Luzon	llocos Region (I)	0	1	0	0
	Cagayan Valley (II)	2	0	0	0
	Central Luzon (III)	1	4	1	0
	Calabarzon (IVA)	4	3	1	1
	Mimaropa Region (XVII)	2	2	1	1
	Bicol Region (V)	0	0	1	0
	CAR (XV)	3	4	1	0
	NCR (XIV)	1	2	3	0
Visayas	Western Visayas (VI)	0	0	3	1
	Central Visayas (VII)	0	0	1	0
	Eastern Visayas (VIII)	1	1	0	0
Mindanao	Zamboanga Peninsula (IX)	0	0	0	1
	Northern Mindanao (X)	0	0	0	0
	Davao Region (XI)	3	0	2	0
	SOCCSKSARGEN (XII)	1	0	2	1
	Caraga Region (XIII)	2	0	3	3
	ARMM (XVI)	0	0	1	0

^£^ Only the pathways with significant Bayes factor support (>3) are shown. For more details on the individual dispersal pathways, please refer to Appendix A. ARMM = Autonomous Region in Muslim Mindanao; CAR = Cordillera Administrative Region; NCR = National Capital Region.

## Data Availability

The *E* gene data has been deposited in the nucleotide database of the National Centre for Biotechnology Information (NCBI) under accession numbers OQ579176-OQ579552.

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
