# Peer review of "Genetic Diversity and Dispersal of DENGUE Virus among Three Main Island Groups of the Philippines during 2015–2017"

_viruses, 2023, doi:10.3390/v15051079_

Round 1

Reviewer 1 Report

The study presented by the authors is very interesting and provides valuable data on the molecular epidemiology of dengue in the Philippines.

Introduction :

In the introduction I think a few lines should be added showing the situation of dengue in the Asian continent and how the Philippines stands in comparison with the other countries of the continent.

The materials and methods are well explained.

I think the results should be presented separately from the discussion, as it is currently it makes it difficult to understand the results and impoverishes the discussion.  The article would be improved if they were presented separately.

Author Response

Response to comments from Reviewer 1

We thank the reviewer for comments to improve the manuscript and its readability. We have revised the manuscript accordingly and have included additional references to support our observations and discussion points.

The study presented by the authors is very interesting and provides valuable data on the molecular epidemiology of dengue in the Philippines.

Introduction :

In the introduction I think a few lines should be added showing the situation of dengue in the Asian continent and how the Philippines stands in comparison with the other countries of the continent.

We have revised the introduction to include additional information on the dengue burden in the Philippines as compared to other countries in the region (Lines 41-59). We also added relevant references to the text.

The materials and methods are well explained.

We thank the reviewer for the comment.

I think the results should be presented separately from the discussion, as it is currently it makes it difficult to understand the results and impoverishes the discussion.  The article would be improved if they were presented separately.

We split the results and discussion in the revised manuscript.

Reviewer 2 Report

The authors present a comprehensive study on the genetic diversity and distribution of an important virus of public health concern. The study is well structured and the manuscript is clearly written. The methods are well described. I would recommend an improvement on the image quality of the figure 1 as the text here is somewhat blurry. Also, there is a section on analysis of selection pressures and although the is table in the supplementary material outlining the outcomes, this does not appear to be adequately discussed within the manuscript text and should be incorporated in some form.

Author Response

Response to comments from reviewer 2

We thank the reviewer for comments to improve the manuscript and its readability. We have revised the manuscript accordingly and have included additional references to support our observations and discussion points.

The authors present a comprehensive study on the genetic diversity and distribution of an important virus of public health concern. The study is well structured and the manuscript is clearly written. The methods are well described. I would recommend an improvement on the image quality of the figure 1 as the text here is somewhat blurry.

We thank the reviewer for the comment. We have provided a high-resolution image of Figure 1 separately.

Also, there is a section on analysis of selection pressures and although the is table in the supplementary material outlining the outcomes, this does not appear to be adequately discussed within the manuscript text and should be incorporated in some form.

We have revised the discussion on selection pressure analysis findings (Lines 352-361).

Round 2

Reviewer 1 Report

I have seen that the authors have made the suggested corrections, so I accept this new version of the manuscript. Congratulations to the authors